# Study of Structural and Optoelectronic Properties of Thin Films Made of a Few Layered WS_2_ Flakes

**DOI:** 10.3390/ma13235315

**Published:** 2020-11-24

**Authors:** Anna Łapińska, Michał Kuźniewicz, Arkadiusz P. Gertych, Karolina Czerniak-Łosiewicz, Klaudia Żerańska-Chudek, Anna Wróblewska, Michał Świniarski, Anna Dużyńska, Jarosław Judek, Mariusz Zdrojek

**Affiliations:** Faculty of Physics, Warsaw University of Technology, Koszykowa 75, 00-662 Warsaw, Poland; michal.kuzniewicz.stud@pw.edu.pl (M.K.); Arkadiusz.Gertych@pw.edu.pl (A.P.G.); karolina.czerniak@pw.edu.pl (K.C.-Ł.); Klaudia.Zeranska@pw.edu.pl (K.Ż.-C.); anna.wroblewska@pw.edu.pl (A.W.); Michal.Swiniarski@pw.edu.pl (M.Ś.); anna.duzynska@pw.edu.pl (A.D.); jaroslaw.judek@pw.edu.pl (J.J.); mariusz.zdrojek@pw.edu.pl (M.Z.)

**Keywords:** WS_2_, tungsten disulfide, thin film, Transition Metal Dichalcogenides (TMDC), Raman spectroscopy, electrical properties, optical properties, thermal properties, first order temperature coefficient, 2D materials

## Abstract

We report a surfactant-free exfoliation method of WS_2_ flakes combined with a vacuum filtration method to fabricate thin (<50 nm) WS_2_ films, that can be transferred on any arbitrary substrate. Films are composed of thin (<4 nm) single flakes, forming a large size uniform film, verified by AFM and SEM. Using statistical phonons investigation, we demonstrate structural quality and uniformity of the film sample and we provide first-order temperature coefficient χ, which shows linear dependence over 300–450 K temperature range. Electrical measurements show film sheet resistance R_S_ = 48 MΩ/□ and also reveal two energy band gaps related to the intrinsic architecture of the thin film. Finally, we show that optical transmission/absorption is rich above the bandgap exhibiting several excitonic resonances, and nearly feature-less below the bandgap.

## 1. Introduction

The group of 2D materials is wide and consists of materials with diverse properties. That diversity enables 2D materials to be used in various applications like nanoelectronics, energy storage, or photonics [1,2,3,4]. Transition Metals Dichalcogenides (TMDs) are an example of such 2D materials, including tungsten disulfide (WS_2_), the most studied representative of TMDs. WS_2_ is a semiconductor with a tuneable direct-indirect bandgap varying from 1.3 to 2.1 eV, for bulk form and monolayer, respectively [5,6]. The bulk form of WS_2_ is popular in the industry due to its lubricanting properties, whereas the 2D counterpart of WS_2_ is known for e.g., interesting electronic and optical properties [7], including a strong Kerr effect, third-order non-linear optical response, or giant two-photon absorption [8]. Its electron mobility has been reported in previous studies and equals 50 ± 7 cm^2^V^−1^s^−1^, ON/OFF ratio is usually around 10^6^ [9,10,11,12]. Thus, WS_2_ is currently a material of great interest to the scientific world, as it is a very promising building block for future beyond-silicon optoelectronics. However, the mentioned properties of WS_2_ have been reported only for individual mono- and few layers fabricated via mechanical exfoliation or chemical vapour deposition. It would be particularly interesting to verify whether, and to what extent those properties can be inherited by more complex systems made of WS_2_ flakes—thin films. The first step to achieve this is to produce a large quantity of flakes and form thin films out of it. For this purpose, liquid-phase exfoliation (LPE) technique can be used. This technique can be modified in many ways and uses various solvents and equipment. Originally, ultrasonic agitation with *N*-methyl-2-pyrrolidone (NMP) had been utilised in the LPE method [13]. So far, a lot of different solvents had been used as a modification of LPE method for WS_2_ exfoliation, e.g., DMF (dimethyloformamide), aqueous PVA (polyvinyl alcohol), water and ethanol, acetone and isopropanol, isopropanol and water, SDS (sodium dodecyl sulfate), etc. [14,15,16,17,18,19,20].

Until now, the NMP has been the most frequently used solvent for the LPE method. It provides a high concentration of exfoliated flakes and a decent shelf-life, but on the other hand, it is an expensive and highly toxic diluent, limiting possible applications outside laboratories [21,22]. Hence, safer solvents are under consideration, e.g., mixture of DI (deionized) water and IPA (isopropyl alcohol) [18]. Flakes exfoliated using LPE techniques can be further employed in the vacuum filtration method to form different kinds of films [23,24,25,26].

Literature provides reports concerning thick free-standing films or heterostructures based on WS_2_ [13,23,24]. For instance, Lu et al. reports thin heterostructures made of graphene and WS_2_ films [24]. The work shows mostly nonlinear optical characteristics of relatively thin layered sandwiches with different thickness (from 60 to 135 nm). On the other hand, Coleman et al. [13,23] demonstrate the production of thick films (50 µm thick) made of WS_2_ and MoS_2_, and hybrid films (200 nm thick) made of WS_2_ and Single Walled Carbon Nanotubes (SWCNT). Reported samples show significant enhancement of over 500 times, compared to disordered WS_2_ films, of thermoelectric properties [25]. Up to now, there were no reports about stable, thin, and transferable WS_2_ films.

Here, we demonstrate how to fabricate very thin WS_2_ films using the liquid-phase exfoliation method without employing harsh chemical reagents and how to transfer it to any arbitrary substrate. The films are composed of thin single flakes (few nanometres thick), forming large size uniform layer, as verified by AFM and SEM. Statistical phonons investigation confirms structural quality and uniformity of the film sample. The temperature-dependent Raman study provides first-order temperature coefficient χ, which is linear over the whole studied temperature range. Moreover, we study the film sheet resistance and also temperature-dependent electrical properties revealing two energy bandgaps related to the intrinsic architecture of the thin film. Finally, we show that optical transmission/absorption above the bandgap exhibits several excitonic resonances, while it is nearly feature-less below the bandgap. 

## 2. Materials and Methods

Si/SiO_2_ and microscope slides, cut in ~1 cm^2^ pieces were used as substrates in this work. The Si/SiO_2_ wafers were 500–550 µm thick, with silicon oxide thickness of 285 ± 30 nm and were N type antimony doped.

Atomic Force Microscopy (AFM, Brucker Icon, Billerica, MA, USA) has been used for structural characterization–topography, morphology, individual flakes, and film thickness. Electron microscopy (SEM, eLine Plus, GmbH, Dortmund, Germany) has been used for defining the continuity, purity, homogeneity of the fabricated thin film. 

Raman study including individual, statistical measurements, and advance thermal characterization had been conducted using Renishaw inVia Raman Spectrometer, Wotton-under-Edge, Gloucestershire, UK. All Raman experiments had been performed in backscattering configuration, in ambient conditions, using green laser wavelength (λ = 532 nm) with low power density in order to avoid damaging the WS_2_ flakes and to prevent additional heating. Temperature-dependent measurement has been conducted in the 300–460 K range. Detailed parameters of each peak were extracted using the Lorentz curve fitting using Marquardt–Levenberg algorithm.

The electrical characterization was carried out using 4 probe measurements in van der Pauw configuration. The four palladium contacts were thermally evaporated on the sample using a mechanical mask technique. The four-probe measurements have been done using Keithley 2450 Source Measure Unit as a current source, Keithley 2182A nanovoltmeter and Keithley 7001 Switch System for proper system configuration control. The sample was placed in the cryostat MicrostatHe2 and Mercury intelligent temperature controller (iTC) system, Oxford, UK for temperature measurements in vacuum conditions (*p*~10^−5^ mbar). The temperature range from 330 K to 440 K was limited due to setup limitations. For proper evaluation of the 4 probe resistances for each contact configuration, we made an I–V characteristic to get the value of slope (resistance).

The Ultraviolet-Visible (UV-Vis) optical measurements (transmittance and reflectance) were made using a photovoltaic (PV) response analyser (PVE300 Bentham, Berkshire, UK) in the range of 300–1700 nm.

## 3. Results and Discussion

The starting point for thin WS_2_ films fabrication is the preparation of the bulk powder suspension and the exfoliation in liquid phase down to individual flakes. Suspensions containing WS_2_ were fabricated using non-harmful, relatively cheap solvents as DI water, and IPA. Commercially available bulk WS_2_ powder purchased from American Elements was used as a precursor in the exfoliation process. In the beginning, a ~0.1 g of 2D micropowder was added to the DI/IPA mixture, previously mixed in the ratio of 7:3. This ratio provides surface tension of approximate value of 40 mN/m, which is similar to NMP’s surface tension [26]. The prepared mixture was placed in an ultrasound bath at maximum power (320 W) for 3 h. After each hour, water in the sonication bath was changed to prevent overheating of the suspension. Then the suspension was centrifuged to remove unexfoliated 2D crystals at 2500 rpm for 75 min. The supernatant was collected and saved for thin-film fabrication. The example of prepared suspension is shown in Figure 1a.

Further, the thin WS_2_ films were produced directly from the exfoliated flakes suspension employing a vacuum filtration method which was conducted as explained below. A 15 mL of suspension was filtered through the cellulose (Millipore) filter with the pore sizes of 25 nm. After WS_2_ film formation on the filter, the setup had been left under vacuum for 2 h to dry off the film and next the vacuum was switched off and left overnight to complete the drying process. Next, the dry film was transferred from the filter to the substrate (any substrate could be used provided that it is resistant to the solvent used during transfer). Here, a Si/SiO_2_ (285 nm thickness of SiO_2_) and glass substrates have been used. The procedure of transferring film from cellulose filter is adapted and optimized from the commonly used method for Carbon Nanotubes (CNT) films [27]. In detail, the film on the filter was gently immersed into isopropyl alcohol and then cut to smaller pieces (5 mm × 5 mm). Each piece of the filter was placed film-side down against the desired substrate and immediately suspended over a bath of boiling acetone (heated to 75 °C) for two hours. This procedure allowed the gentle dissolution of the cellulose filter. Afterwards, the samples were immersed several times in a pure non-heated acetone bath to remove the residual filter from the samples. Finally, samples are placed into an isopropanol bath for a few minutes and then dried with compressed nitrogen.

We note that the above-described method of production of thin films using DI/IPA based suspension and the possibility to transfer films on different substrates is universal for other 2D materials such as GeS, GeSe, SnSe_2_, hBN, NbSe_2_, MoS_2_ [28], as shown in the Appendix A.

The image of thin-film and SEM scan of its surface are shown in Figure 1b,c, demonstrating the continuity of the film. Based on an AFM scan of individual flakes (see Appendix A) distribution of flake’s thickness was determined, which is provided in Figure 1d. The average flake thickness is estimated to ~3.9 (±1.5) nm. Accordingly, also to AFM, the roughness parameter of the whole film was estimated to be ~7 nm (Appendix A) and the averaged thickness of the whole film was measured to be ~45 nm. The film thickness had been evaluated by measuring the AFM profile on the edge of the film (see Appendix A).

The structural properties of WS_2_ film were verified using Raman spectroscopy. Figure 2a presents a typical single Raman spectrum of WS_2_ thin film (on SiO_2_/Si substrate) exhibiting two main peaks: E^1^_2g_ (~353 cm^−1^) and A_1g_ (~421 cm^−1^) accompanied by several lower energy peaks. The spectrum is consistent with other spectra obtained for WS_2_ flakes [29,30]. Statistical analysis of main phonon position distributions is further performed using the Raman mapping approach, where data is collected within 1 mm × 1 mm spatial map with step: 4 μm, delivering 251 × 251 single plots (see Figure 2b). Data presented in a form of histograms of peak position (phonon energy) clearly shows the structural uniformity of main Raman modes position distribution (full width at half maximum(FWHM) < 0.3 cm^−1^ in large areas of the WS_2_ film. The fluctuation of modes position is relatively low and mainly depends on the different thickness of the individual flakes distribution. In addition, the averaged difference between two main modes (A_1g_–E^1^_2g_) is ~68 cm^−1^, suggesting that individual flakes thickness corresponds to several layers (~5–6) [31], which is also consistent with AFM study [32] (see Figure 1d).

Next, the temperature-dependent Raman analysis was performed. Figure 2c shows the evolution of Raman spectra with increasing temperature in the 300–460 K range. The main Raman modes E^1^_2g_ and A_1g_ downshift with increasing temperature by 1.39 and 0.9 cm^−1^ respectively, which is an expected behaviour for any type of 2D materials [33,34,35]. To quantify the described effect, one can derive first-order temperature coefficient according to the following equation: $\omega \left(T\right)=\omega_{0}+\chi T$, where: *χ*—is the first order temperature coefficient, $\omega_{0}$—phonon frequency at 0 K. Based on this equation the extracted χ coefficient values: 0.00732 cm^−1^/K for mode E^1^_2g_ and 0.00984 cm^−1^/K for mode A_1g_. Interestingly, these values are approximately up to two times lower than for supported and suspended single WS_2_ crystals made by chemical vapor deposition (CVD) [6]. The lower value of χ likely stems from a large number of interfaces between flakes within the film that could significantly lower the thermal dissipation properties of the whole film. The values of χ for WS_2_ film could be useful for the determination of the thermal conductivity of the thin film [6].

The electrical measurements have been performed using the 4 probe method (see Figure 3a showing the device used in our study), which ensures that there is no contribution of contacts and wiring resistance on the sample’s sheet resistance (R_S_). Measurements took place in the temperature range of 300 K–440 K. Figure 3b illustrates current-voltage (I-V) characteristics in a form of hysteresis proving that measurements are repeatable. The origin of the hysteresis stem from a large number of interfaces between flakes in the film and film-metal contact interface that could lead to some charging effects. We found that the resistance of WS_2_ thin film is relatively high (R_S_ = 48 MΩ/□ for T = 440 K), but not significantly different than previously reported [36]. Figure 3c shows the natural logarithm of the R_S_ versus (k_B_T)^−1^, where the slope of the linear dependence directly represents the value of the activation energy. One can see that we have obtained two linear regions, which are represented by two activation energies of E_g1_ = 1.8 eV, and E_g2_ = 0.38 eV. This suggests that electronic transport in such disordered film has two contributions: (i) the intrinsic properties of the WS_2_ (E_g1_) and (ii) the thin potential barriers between the flakes (E_g2_), which are in contrast to previously investigated tungsten disulfide thin films [37,38].

Finally, the thin WS_2_ (transferred on quartz substrate) were characterized using broadband UV-VIS spectroscopy. The corresponding data are shown in Figure 4 including the comparison of the optical absorption (A), reflectance (R), and transmittance (T) for WS_2_ film. All plots above ~700 nm (1.77 eV) are rather featureless except slow monotonic increase/decrease of the T and R values, respectively, which is clearly related to the bandgap of the WS_2_ flakes. This is also in agreement with the electrical measurements shown in Figure 3c.

Interesting features in the spectra can be observed below 700 nm. Absorption spectrum (here derived in Figure 4) contains several absorption resonances at ~623, 510, 448 and 383 nm. These features with similar values are also seen for individual WS_2_ few-layer crystals [39]. These peaks are also reflected in the transmittance spectra. Interestingly, they might be related to excitonic resonances and characteristic A, B, C, and D excitons, also observed for few-layer WS_2_ flakes reported in the literature [5,40], indicating the origin of the peaks as direct gap transition at K point considering the A and B peaks and for the C peak as the optical transitions of valence and conduction bands.

## 4. Conclusions

In conclusion, we have established the methodology for surfactant-free production of thin films made of 2D flakes and transferable on any arbitrary substrate, here using tungsten disulfide WS_2_ as an example. Using temperature-dependent phonon and electrical properties assisted by AFM/SEM study, we demonstrated the structural quality and architectural uniformity of the WS_2_ film over a large scale. Finally, we showed that the optical properties of the film are similar to those observed in WS_2_ monolayers indicating the bandgap and exhibiting several excitonic resonances and are consistent with electrical measurements. The results here give us a better insight into the properties of the new thin film structure, that could pave for new, potential application, especially in field of electronics, optoelectronics, heat management or thermoelectricity.

## Figures and Tables

**Figure 1 materials-13-05315-f001:**
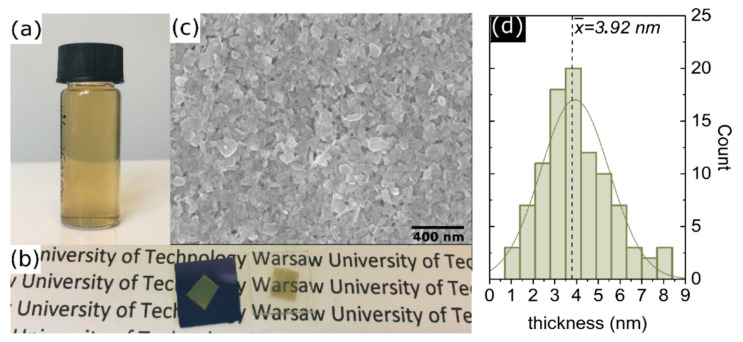
(**a**) Suspension of tungsten disulfide made via “green” liquid exfoliation, (**b**) thin films of WS_2_ fabricated using vacuum filtration on Si/SiO_2_ and glass substrates, (**c**) SEM scan of fabricated thin films, (**d**) flakes thickness distribution, based on AFM study.

**Figure 2 materials-13-05315-f002:**
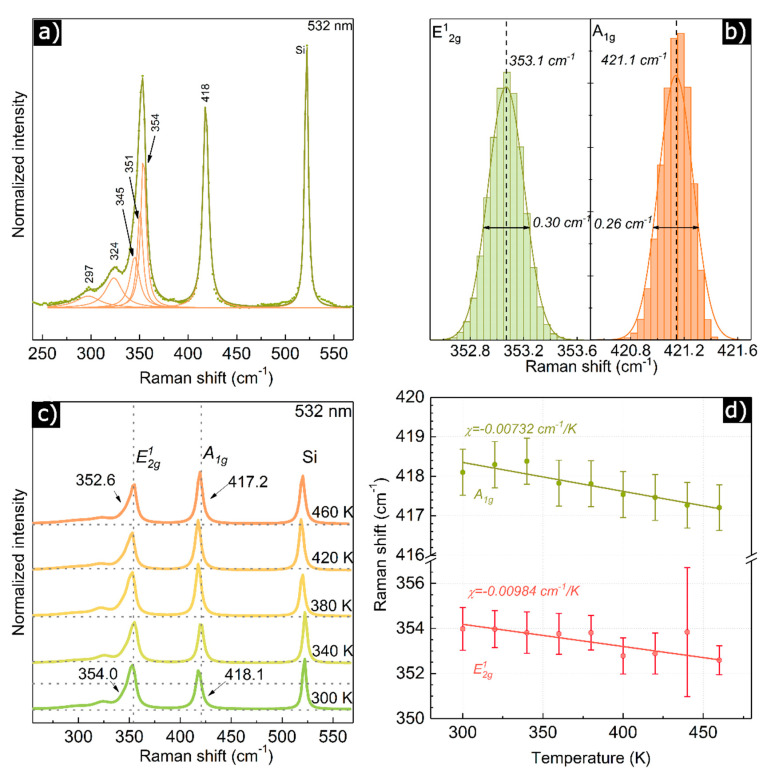
(**a**) A typical single Raman spectrum obtained for WS_2_, laser wavelength: λ = 532 nm, RT (room temperature); (**b**) statistical distribution of two main Raman modes in WS_2_ film. The average mode positions and standard deviation are as follow: $\bar{x}$ = 421.14 cm^−1^,σ = 0.13 cm^−1^ for the A_1g_ mode, $\bar{x}$ = 353.07 cm^−1^, σ = 0.13 cm^−1^ for the E^1^_2g_ mode; (**c**) temperature-dependent Raman spectra, (**d**) detailed temperature analysis for two main modes in Raman spectrum for WS_2_.

**Figure 3 materials-13-05315-f003:**
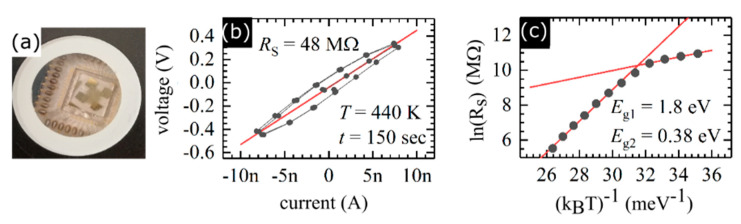
(**a**) Photography of device used for electric measurements, (**b**) I–V characteristic obtained for 440 K, t = 150 s is waiting time before collecting results, the red solid line represents linear fitting performed for the resistance evaluation (slope) (**c**) Arrhenius dependence with extracted bandgap value.

**Figure 4 materials-13-05315-f004:**
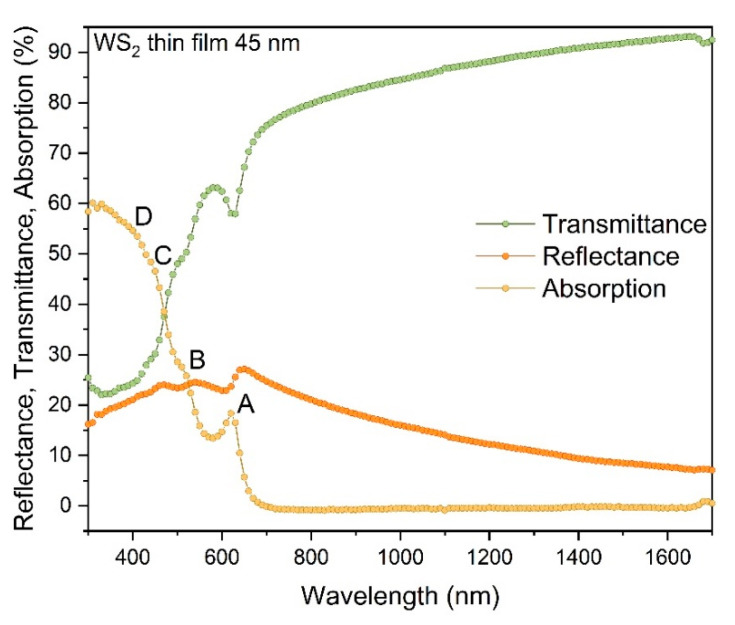
Optical characteristic of WS_2_ thin film: absorption (A), reflectance (R), transmittance (T).

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
