# Peer review of "Study of Structural and Optoelectronic Properties of Thin Films Made of a Few Layered WS2 Flakes"

_materials, 2020, doi:10.3390/ma13235315_

Round 1
Reviewer 1 Report
This paper provides an easy and cost effective alternative method of producing thin films made from WS2 flakes. This work is a good idea, with decent electrical and material characterisation. However, I do have some concerns on some things in this work that the authors need to address for their work to be of the appropriate level to be published in Materials MDPI journal.
- ‘Thus, WS2 is currently of great interest as it is a very promising building block for future beyond-silicon optoelectronics. However, the mentioned properties of WS2 have been reported only for individual mono- and few layers fabricated via mechanical exfoliation or chemical vapour deposition.”
- ‘Up to date, the NMP is in the most frequently used solvent for the LPE method. It provides a high concentration of exfoliated flakes and a decent shelf-life, but on the other hand, it is an expensive and highly toxic diluent, limiting the application outside the laboratories.’
iii. ‘Flakes exfoliated using LPE techniques can be further employed in the vacuum filtration method to form different kinds of films.’
Reference(s) missing here.
- Can you provide some information on the mode for AFM measurements as well as the tips used? You do not mention how you translate the information that you use. How you calculate rms values. How you calculate flake thickness (and what you mean by that).
- The same, can you provide more information on SEM measurements. What are the conditions of the measurement? Any sputtering above?
- I would refrain from using eco-friendly for IPA. It is not considered a ‘green’ solvent.
- When describing the method of production words like ‘appropriate’ ‘should’ ‘approximate’ and anything similar do not provide the certainty of information needed. Please rephrase these using words that show accuracy and control of the process.
- Give information in the materials and methods on the substrates you use. What kind of Si? Is it a wafer? What characteristics? What kind of glass? Borosilicate? Microscope slide?
- ‘A certain amount of suspension’. How much is this certain amount? If it is not the same, why it is not the same every time and what is the range of amounts used for this work?
- Line 122: Typo ‘Afterward’ is afterwards.
- I am completely lost on how you calculate the thickness of the flakes. The measurements in the supplementary information suggests an rms value of ~7.5. This means that your roughness is ~50-60 nm in height. You need to provide the measurement of a 2x2 μm (or less) area to be able to identify the boundaries of the flakes. Flake thickness would be more possible to identify if you can distinquish them in the image and from what we see in the AFM images, this is not the case. The area you choose is really big to be able to claim that. (20x20 μm and 50x50 μm if I am not mistaken). The information received by these images are dependent on the flakes that create this ‘roughness’ but still the outcome of the analysis using this data is questionable.
- Line 130. SEM image is only image c not b.
Reviewer 2 Report
Two-dimensional materials, such as tungsten disulfide flakes, are promising for use in next-generation non-silicon electronics, optoelectronics, energy harvesting etc. Authors present a new eco-friendly fabrication method of WS2 flakes and thin films composed of such flakes and characterize them by different techniques. However, some of the observed characteristics are explained scarcely or not explained in the manuscript text.
I recommend this manuscript to be published in Materials with the minor revisions according to the following remarks:
- Please explain in more details how the individual flake thicknesses were obtained from the AFM measurements. Probably the corresponding film surface profile presented in the Supplementary information could help. Alternatively, make a reference to the known method.
- Please explain the fact that the A1g mode position values shown in Fig. 2c and Fig.2d are well below the statistical distribution of that parameter over the large WS2 film area (Fig. 2b).
- Could be explained (optionally) the appearance of a hysteresis in I-V measurements (Fig. 3b)?
Reviewer 3 Report
The authors propose in this paper a green method to exfoliate WS2 flaxes and transfer them as a film, to different substrates, in this case oxidized silicon.
The method is exposed in details and supported by different characterization techniques, AFM, Raman, Optical and electrical. Conclusions are well supported by the results.
Some minor questions have to be addressed:
in Figure 1b the superposition of the sample picture with the “university of Technology Warsaw” affiliation makes the photo itself less clear, i suggest to put a white background, with at least the affiliation on one side of the photo.
At row 194 the sentence stars with (here derived as…..and does not have and end, please check.
In the Abstract, at row 17 there is a symbol missing
Reviewer 4 Report
A interesting paper with many aspects very well written and a topic that has relevance.
Some key improvements required to bring this up to publication standard however to within the guidelines given.
References need to be in Arabic number format, not roman. In places these are superscript and others not.
Section numbering is incorrect - all say section 1.
Figure 1 caption repeats figure 1.
Abbreviations need to be written out in full for each new occurrence, e.g. Carbon Nano Tube (CNT) at lines 56 & 118.
Line 40 - poor English, replace with "Originally, ultrasonic agitation with.."
Line 45 - poor English, consider "Until now"
Line 70 2. materials etc
Line 80 Marquardt not Marquart, after the excellent Donald Marquardt
Line 92 3. results etc
Line 107 duplication of Figure 1
Line 172 - is this meant to be Fig 3.b? Fig 3a seems to be a photo only
Line 193 - poor English , remove the "the" before 700nm
Line 201 4. conclusions
This paper has merit, is based on good science and should be published but I'm left with the feeling that I still don't follow what is novel and yours.
There is not enough detail in the "Conclusions" section. This is a great place to reiterate and reinforce what is novel and potential next steps / improvements.
Round 2
Reviewer 1 Report
The manuscript has been significantly improved. There are some minor issues to be addressed:
- There is still some information on AFM measurements that are missing. Are the images you show here 256x256, 512x512, 1024x1024 (or more?)?? You mention "high resolution scan" but what do you mean by that? Why do you not use a 2x2 or 4x4 μm scan size so that you extract more detailed information and validate the height you have calculated using the method you explain in the supplementary information?
- Some syntax and grammatical error in the document, especially in the newly added sentences. Please revise.
